# SALL Proteins; Common and Antagonistic Roles in Cancer

**DOI:** 10.3390/cancers13246292

**Published:** 2021-12-15

**Authors:** Claudia Álvarez, Aracelly Quiroz, Diego Benítez-Riquelme, Elizabeth Riffo, Ariel F. Castro, Roxana Pincheira

**Affiliations:** Departamento de Bioquímica y Biología Molecular, Facultad de Ciencias Biológicas, Universidad de Concepción, Concepción 3349001, Chile; clalvarez@udec.cl (C.Á.); aracellyquiroz@udec.cl (A.Q.); diegobenitez@udec.cl (D.B.-R.); elizriffo@udec.cl (E.R.); arcastro@udec.cl (A.F.C.)

**Keywords:** SALL1, SALL2, SALL3, SALL4, cancer, epigenetic regulation, Wnt, PTEN, biomarker

## Abstract

**Simple Summary:**

Transcription factors play essential roles in regulating gene expression, impacting the cell phenotype and function, and in the response of cells to environmental conditions. Alterations in transcription factors, including gene amplification or deletion, point mutations, and expression changes, are implicated in carcinogenesis, cancer progression, metastases, and resistance to cancer treatments. Not surprisingly, transcription factor activity is altered in numerous cancers, representing a unique class of cancer drug targets. This review updates and integrates information on the SALL family of transcription factors, highlighting the synergistic and/or antagonistic functions they perform in various cancer types.

**Abstract:**

SALL proteins are a family of four conserved C2H2 zinc finger transcription factors that play critical roles in organogenesis during embryonic development. They regulate cell proliferation, survival, migration, and stemness; consequently, they are involved in various human genetic disorders and cancer. SALL4 is a well-recognized oncogene; however, SALL1–3 play dual roles depending on the cancer context and stage of the disease. Current reviews of SALLs have focused only on SALL2 or SALL4, lacking an integrated view of the SALL family members in cancer. Here, we update the recent advances of the SALL members in tumor development, cancer progression, and therapy, highlighting the synergistic and/or antagonistic functions they perform in similar cancer contexts. We identified common regulatory mechanisms, targets, and signaling pathways in breast, brain, liver, colon, blood, and HPV-related cancers. In addition, we discuss the potential of the SALL family members as cancer biomarkers and in the cancer cells’ response to therapies. Understanding SALL proteins’ function and relationship will open new cancer biology, clinical research, and therapy perspectives.

## 1. Introduction

SALL proteins are transcription factors that belong to the Spalt-like (Sall) family, broadly conserved through evolution. They are present in nematodes, flies, planarians, bilaterians, and vertebrates. They were first identified in *Drosophila melanogaster*, which harbors two paralogs: *spalt major* (*salm*) and *spalt-related* (*salr*). Both proteins play a role in the homeotic specification of the embryonic termini, wing patterning, sensory organ development, and photoreceptors specification [1,2]. A recent study in bilaterians suggests an ancestral role of *sall* in neural development [3].

Vertebrate genomes harbor four paralogs, *SALL1–SALL4*, apparently originated by several duplication events of the *spalt* locus and evolved from one ancestor more closely related to Drosophila (salm ortholog) [2]. A phylogenetic analysis of SALL proteins indicates that SALL1 and SALL3 derived from one common ancestor, and SALL4 derived from a more distant one. SALL2 shares the least homology, being the most dissimilar member of the SALL family, especially in the *C*-terminal region [4]. They are characterized by multiple zinc finger domains throughout the protein, a glutamine-rich (poly-Q) region important for protein–protein interactions, and a conserved twelve-amino-acid domain at the N-terminal region responsible for the repression activity of SALL proteins, mediated by an interaction with the Nucleosome Remodeling and Deacetylase (NuRD) complex [5] (Figure 1). The zinc finger domain 1 corresponds to the C2HC class, and the rest of the domains (2–5) correspond to the C2H2 zinc fingers arranged in pairs. The second finger from each pair contains a characteristic domain called Sal-box (FTTKGNLK), present in other zinc finger transcription factors such as Schurri and PRDII-BF1 [3]. The third zinc finger domain contains an associated finger also highly conserved among orthologs. The function of SALL proteins requires nuclear localization, likely depending on the zinc finger 1 [5] (Figure 1).

SALL proteins act as tumor suppressors, oncogenes, or have a dual function depending on the tissue, genetic context, epigenetic regulation, and specific SALL isoforms implicated, among others. SALL1, SALL2, and SALL4 cancer-related isoforms have been reported [6,7,8,9,10]. Considering that isoforms differ in structure and might have differential expressions and cellular location, they could be responsible for some of the different functions of SALL proteins in cancer (Figure 1).

Current reviews on SALL proteins’ role in cancer are available; however, they are only focused on SALL4 or SALL2 [4,11,12,13,14,15], lacking an integrated view of the SALL family. Most cancer studies relate SALL4 to an oncogenic function and SALL1–3 to a tumor suppression role; still, there is evidence that SALL1 and SALL2 could play a dual role. Here, we will update information and integrate studies involving the SALL proteins in tumor development, progression, and cancer therapy, highlighting their synergistic and/or antagonistic functions in similar cancer types. We will also discuss their potential as cancer biomarkers and therapeutic targets. Understanding SALL proteins’ function and how they behave in similar cellular contexts will open new perspectives in cancer biology, clinical research, and therapies.

## 2. Essential Roles of *SALL* Genes during Development

Vertebrate SALL proteins participate in the development of extremities and organs, including the brain, kidney, eye, and heart. Accordingly, *SALL* genes are implicated in human genetic disorders [1,2,3,4,5,16]. Mutations of *SALL1* cause the Townes–Brocks syndrome (TBS), a rare autosomal malformation syndrome characterized by anal, renal, limb, and ear anomalies (Reviewed in [1,2]). Similarly, *SALL4* mutations cause the Okihiro/Duane-radial ray syndrome (DRRS), an autosomal dominant condition characterized by upper-limb anomalies, ocular anomalies, and renal anomalies in some cases [17]. Meanwhile, *SALL2* deficiency causes recessive ocular coloboma [18]. *SALL3* deficiency is associated with ocular anomalies and facial dysmorphism of the human 18q deletion syndrome [19].

Functional studies using knockout mice confirmed the essential roles of *Sall1*, *Sall3*, and *Sall4* during development. Loss of function of these genes results in perinatal or neonatal lethality due to organ alterations during embryonic development. The organ alterations include kidney agenesis or dysgenesis, abnormal cranial nerve morphology, and exencephaly [20,21,22]. Two *Sall2* knockout (KO) models exist; the first *Sall2*KO model did not show an essential role for *Sall2* in embryonic or kidney development [23]. However, the second *Sall2*KO model showed severe neural tube defects and defects in the optic fissure closure, similar to the phenotype of coloboma patients [18,24]. Differences between the phenotype of *Sall2*KO models might be related to the different genetic backgrounds of the mice strains used [23,24]. *Sall2*KO did not show spontaneous tumor formation [23], but when crossed with tumor-susceptible mice *p53*^−/−^, it exhibited significantly accelerated tumorigenesis, tumor progression, and mortality rate among *Sall2^+/+^*/*p53*^−/−^ mice. The *Sall2*^−/−^ or *Sal2*^−/+^/*p53*^−/−^ mice showed thymus T-cell lymphoma that metastasized to the liver, lung, kidney, marrow, peripheral blood, and central nervous system, while in most *Sall2^+/+^*/*p53*^−/−^ mice, the lymphoma was limited to the thymus and adjacent organs such as the lung [25]. Moreover, supporting a tumor suppressor function, the immortalized *Sall2*^−/−^ MEFs showed an enhanced growth rate, foci formation, and anchorage-independent growth in comparison to the immortalized *Sall2*^+/+^ MEFs [26].

In development, common findings on SALL proteins include a direct interaction with chromatin remodeling complexes, such as the SWI/SNF or NuRD complexes, and an association with the canonical Wnt/β-catenin pathway [5,10,27,28,29,30]. Particularly relevant for the SALLs function is their interaction with the NuRD complex. NuRD is involved in global transcriptional repression and the regulation of specific developmental processes [29,31]. SALLs interact with the NuRD complex through the conserved 12-amino-acid motif. This motif is not present in some SALL isoforms (Figure 1), suggesting that NuRD is essential in differentiating their function. However, there is a lack of studies addressing this issue. Most of the studies have focused on the functional relationship of NuRD with SALL1 or SALL4 in development and cancer. In kidney development and leukemogenesis, the function of SALL4 through the NuRD complex relies on the repression of PTEN and SALL1 [32]. In other contexts, the interaction between SALL4 and NuRD impacts different genes. SALL4 was involved in spermatogonial differentiation. SALL4/NuRD repressed the expression of the tumor suppressor genes *Foxl1* and *Dusp4*, associating SALL4 function with the maintenance of undifferentiated spermatogonial activity and stem cell-driven regeneration [33]. However, in the context of pluripotent cell transcriptional programs, free SALL4 regulates transcription independently of NuRD [34].

The SALL1/NuRD complex is also involved in kidney development, inhibiting the premature differentiation of nephron progenitor cells. The disruption of SALL1/NuRD interaction in vivo resulted in the accelerated differentiation of nephron progenitors and bilateral renal hypoplasia [28]. In Xenopus embryos, SALL1 interaction with NuRD directly repressed Gbx2, a transcription factor for cell pluripotency and differentiation in the embryo [35]. Interestingly, SALL1 association with NuRD is disrupted by the protein kinase C phosphorylation at serine 2 of the repression motif, suggesting that this kinase regulates the NuRD-dependent repression function of SALL1 [35]. SALL1 phosphorylation by PKC may also be involved in breast cancer [36]. SALL1/NuRD inhibited breast cancer cell growth, proliferation, and metastasis, and a phosphomimetic mutation of SALL1 impaired its tumor suppressor function. Whether Gbx2 is associated with the tumor suppressor function of SALL1 is currently unknown.

As with many genes that play essential roles in organogenesis during embryonic development, *SALL* genes are involved in cancer. Developmental pathways are crucial for the cellular processes required during embryonic stages, such as epithelial-mesenchymal transition (EMT), coordinated migration, and cell proliferation, which are also essential at different stages of tumor progression [37]. Increasing evidence shows an association of SALL proteins with these processes, which are discussed below.

## 3. Common Cellular Functions and Targets of the SALL Proteins in Cancer

The number of identified SALL proteins’ target genes has increased in recent years. They are associated with diverse cellular events such as proliferation, cell death, migration, invasion, and stemness.

### 3.1. Cell Proliferation

Several studies have established the role of SALL proteins in cell proliferation, acting as oncogenes or tumor suppressors under different pathological contexts. For instance, ectopic SALL2 expression inhibited SKOV3 ovarian cancer cell proliferation by a mechanism that involves the positive transcriptional regulation of cell cycle inhibitors such as p21 and p16 [38,39]. Accordingly, SALL2 depletion increased A2780 ovarian carcinoma cell proliferation [40]. The loss of *Sall2* in mouse embryonic fibroblasts (MEF) enhanced cell proliferation and showed faster postmitotic progression through the G1 and S phases. The mechanism is related to the transcriptional derepression of two SALL2 targets, cyclins D1 and E1 [26]. On the contrary, SALL4 accelerated cell cycle progression in cervical, lung, and breast cancer cells, as well as in esophagus squamous cell carcinoma and glioma [41,42,43,44,45]. The opposite roles of SALL2 and SALL4 in cell proliferation agree with the regulation of c-MYC, a transcription factor involved in cell growth and cell cycle control. SALL2 directly binds to the nuclease hypersensitive element in the promoter of *c-MYC*, repressing its expression [46]. However, SALL4 indirectly increases the levels of c-MYC by activating the Wnt pathway. SALL4 enhanced the proliferative capacity of HeLa and SiHa cervical cancer cells through the positive transcriptional regulation of *CTNNB1* [41]. *CTNNB1* encodes β-catenin, a transcriptional cofactor of TCF/LEF in the Wnt signaling pathway. Additionally, SALL4 is directly bound to β-catenin, which activates the Wnt pathway in AML (acute myeloid leukemia) [8]. In both studies, Wnt pathway activation by SALL4 increased c-MYC and cyclin D1, which are related to increased proliferation, survival, EMT, and metastasis [8,41].

SALL1 is also associated with increased β-catenin expression in human primary AML samples, and the inhibition of SALL1 resulted in decreased cell proliferation and AML engraftment in NSG (NOD scid gamma) mice [9]. Interestingly, similar to SALL4, SALL1 interacts with β-catenin in human kidney BOSC23 cells derived from HEK293T cells [47], suggesting a common mechanism for Wnt pathway activation via the interaction of β-catenin with SALL1 and/or SALL4. However, in contrast to the AML context, SALL1 over-expression in MDA breast cancer cells inhibited tumor cell growth and proliferation. It promoted cell cycle arrest by increasing Cyclin A2, Cyclin B1, Cyclin E1, CDK2, and CDK4, which are essential for checkpoint regulation in the G1-S transition and S phases [36]. These findings suggest a dual role for SALL1 in cancer, depending on the cell context.

### 3.2. Apoptosis and Cell Survival

SALL proteins’ target genes are also associated with the regulation of apoptosis, indicating that SALL2 and SALL4 play opposite roles in this process. Using chromatin immunoprecipitation followed by microarray hybridization in the human acute promyelocytic leukemia cell line NB4, Yang and collaborators validated the SALL4 upregulation of anti-apoptotic genes, such as Bmi-1, BCL2, DAD1, TEGT, BIRC7, and BIRC4, and the negative regulation of pro-apoptotic genes, such as TNF, TP53, PTEN, CARD9, CARD11, ATF3, and LTA. Moreover, the inhibition of SALL4 induced apoptosis in NB4 cells, increasing DNA fragmentation as well as caspase-3 and annexin V levels [48]. On the other hand, the apoptotic cell response to genotoxic stress and Trichostatin A (TSA) treatment required SALL2 [49,50,51]. In response to doxorubicin- and etoposide-induced genotoxic stress, SALL2 induced pro-apoptotic genes such as *BAX* and *PMAIP1* (also known as *NOXA*) in human ovarian surface epithelial (HOSE) cells and MEFs. Particularly noteworthy is the pro-apoptotic role of SALL2, which was independent of p53 expression, suggesting the key role of SALL2 in the response of cancer cells to therapy in p53 inactive cancer contexts [49,50].

### 3.3. Cell Migration and Invasion

Migratory and invasive cell capacities increase during tumor development, which are strongly associated with metastasis in advanced stages of cancer. There are several mechanisms by which tumor cells acquire these characteristics of malignancy. One of the central mechanisms is the inhibition of PTEN, a phosphatase that blocks the PI3K signaling pathway, inhibiting cell migration, proliferation, and survival [52]. SALL4 repressed PTEN expression through its interaction with the NuRD complex and favored the development of AML in mice [32]. In ICC-9810 cholangiocarcinoma cells, SALL4 inhibited migratory and invasive capacities through the repression of PTEN and the upregulation of Bmi-1 [53]. Similarly, SALL1 inhibition increased PTEN expression in AML cell lines and primary samples and downregulated mTOR, β-catenin, and NF-қB expression [9]. SALL1 is bound to the NuRD complex in breast cancer; thus, it is likely that SALL1 and SALL4 share a similar repressive mechanism for PTEN regulation. However, no changes in PTEN expression were detected in breast cancer cells with SALL1 over-expression, suggesting that the regulation of PTEN by SALL1 is tissue-specific [36].

Meanwhile, SALL2 induces PTEN expression. In breast cancer cells, SALL2 silencing activated the AKT/mTOR pathway via the downregulation of PTEN. The mechanism involves the positive regulation of PTEN through the direct binding of SALL2 to canonical GC-rich consensus elements in the PTEN promoter [54]. Although this study did not associate SALL2-dependent PTEN regulation with cell migration, previous studies demonstrated that SALL2 expression correlated with impaired cell migration in human ovarian and esophageal carcinoma cell lines [40,55]. Additionally, the *CDH1* and *VIM* genes, involved in migration, invasion, and EMT, are common targets for SALL1 and SALL4 in breast cancer (discussed below).

### 3.4. Stemmess

Maintenance of stemness is another essential feature of the heterogeneous cell population within the tumor, increasing its complexity by conferring the ability to differentiate into many unrelated cell types.

The role of SALL proteins in stemness maintenance is relevant during embryo development and cell fate. For instance, SALL2 and SALL4 are necessary factors for the self-renewal of hematopoietic stem cells (HSC) [25,56,57]. Interestingly, several studies suggest that SALL1 and SALL4 act as stemness or differentiation factors, depending on the development stage and the cell type involved. SALL1 is required for the stem cell maintenance of kidney, heart, and spermatogonial progenitors [27,58,59,60]. However, SALL1 also participates in the heart and odontoblast lineage differentiation [59,61]. Similarly, SALL4 plays opposite roles in postnatal spermatogenesis and embryonic germ cells [62]. During spermatogonia differentiation, SALL4 sequesters Plzf, a factor required to maintain adult stemness. This interaction leads to the expression of the differentiation marker KIT and the repression of SALL1 [62]. SALL1 expression in the germline is specific for spermatogonia progenitor cells. It was proposed as one of the factors involved in spermatogonial stem cell self-renewal [58].

SALL4 was proposed as a crucial factor for maintaining pluripotency in embryonic stem cells (ESCs) by its direct interaction with the core master regulators SOX2 and OCT4 [63]. Recent research has revealed that SALL4 maintains the pluripotent state in ESCs by regulating a set of AT-rich genes that promote neuronal differentiation. Worthy of note here is that the AT-rich gene pull-down by SALL4 depends on the C2H2 zinc-finger cluster 4 (ZFC4) domain, also found in SALL1 and SALL3, but not in SALL2 [64].

Interestingly, the putative tumor suppressor SALL2 was identified as one of the critical transcription factors necessary for maintaining the tumor propagating cells in glioblastoma. SALL2 interacted with SOX2, OCT4, and Nanog in this specific context, promoting stemness and aggressive behavior [65]. Similarly, SALL1 can interact with SOX2 and Nanog, but not with OCT4, and consequently induce an undifferentiated state. SALL1 also suppresses ectodermal and mesodermal differentiation. Meanwhile, SALL1 overexpression was found to inhibit the induction of gastrulation markers (T brachyury, Goosecoid, and Dkk1) and neuroectodermal markers (Otx2 and Hand1) [66]. Recently, SALL3 was identified as part of a small set of transcription factors, including SOX2 and SALL2, that interact with the Mediator complex in neural stem cells [67]. Altogether, these studies identified SOX2 as a common SALL protein partner, relevant for the maintenance of stemness.

## 4. Common Regulatory Mechanisms for SALL Proteins in Cancer

The regulation of SALL proteins is an open field of study and involves several different mechanisms for each family member. These include genetic alterations and specific transcriptional, posttranscriptional, and posttranslational regulation. However, a common regulatory mechanism for all family members relates to epigenetic modifications, including chromatin modifications and microRNAs (miRNAs).

The loss of heterozygosity (LOH) was reported in overlapping regions of *SALL* genes in several independent cancer studies and was associated with poor prognosis and metastatic recurrence. These regions include *SALL1* (16q12.1) [68,69], *SALL2* (14q11.1–12) [70,71,72,73,74,75,76], and *SALL3* (18q23) [76,77,78]. On the other hand, chromosomal amplifications were found in the *SALL4* region (20q13.2) [79,80,81,82,83,84,85].

Specific transcription factors regulate the expression of SALL genes. SALL2 is transcriptionally activated by AP4 and Sp1 [51,86] and repressed by WT1, p53, and FosL1 [87,88,89]. On the other hand, TCF/LEF, STAT3, and CDX1 are transcriptional activators of SALL4 [10,90,91] (Figure 2). Remarkably, SALL4 controls their expression and represses SALL1 and SALL3, thus regulating the stemness of ES cells [63]. In murine transgenic models, SALL4 represses SALL1 and PTEN through the NuRD repressor complex, leading to pathologies such as cystic kidney and myeloid leukemia, respectively [32].

Promoter hypermethylation is frequent for *SALL1*, *SALL2*, and *SALL3*, and the regulation of 3′UTR by miRNAs appears as a typical regulatory mechanism for *SALL4*. Epigenetic modifications on *SALL* genes are consistent with their prominent roles as tumor suppressors or oncogenes. For instance, associated with their tumor suppressor role, the hypermethylation of the *SALL1* and *SALL2* promoters were described in breast cancer and esophageal squamous cell carcinoma (ESCC) [54,55,92,93]. The hypermethylation of the *SALL2* promoter was associated with aggressive and tamoxifen-resistant breast cancer phenotypes [54]. In oral squamous cell carcinoma (OSCC), *SALL2* promoter hypermethylation positively correlates with *SALL1* and *SALL3* promoter methylation status and aggressive tumor behavior [94].

The *SALL* promoters are also aberrantly methylated in HPV-related cancers. Several studies indicate that the hypermethylation of *SALL1* and *SALL3* promoters correlates with poor outcomes and recurrence in head and neck squamous cell carcinoma (HNSCC) [95,96,97]. However, SALL4 is upregulated in this type of cancer, and its expression correlates with disease recurrence and decreased disease-free survival. High SALL4 expression positively correlated with DNA methyltransferase 3 alpha (DNMT3A) expression and the increased methylation rate of 11 tumor suppressor genes. Still, there was no significant correlation between SALL4 expression and *SALL1*, *SALL2*, and *SALL3* methylation status [121]. Aberrant hypomethylation of the *SALL4* promoter is described as a common event in AML and myelodysplastic syndrome (MDS) [98,99].

Table 1 shows the repertory of miRNAs regulating SALL1, SALL2, and SALL4 in different cancer types. Most of them target SALL4, and some miRNAs are common among cancers. More specifically for SALL4, miR-16 is common in glioma and gastric cancer, miR-103 in glioma and oral squamous cell carcinoma, and miR-107 in glioma and osteosarcoma. For more comprehensive information on SALL4 and miRNAs and the strategies targeting the miR/SALL4 axis in cancer, see a recent review by Liu J and collaborators [100].

These studies identified common upstream regulation, targets, and cellular functions. However, their effects seem to be opposite or synergic, depending on the tumor context and the implicated SALL protein. Concerning the role of SALL3 in cancer development and progression, the available information is still scarce, but we will discuss recent studies on SALL3 by cancer subtype in the next section. In Figure 2, we illustrate SALL proteins’ shared targets, partners, cellular functions, and regulatory mechanisms.

## 5. SALL Proteins in Cancer

SALL proteins are altered in various cancer types (Table 2 and Figure 3). Alterations include deregulation in gene expression, isoform expression, and genetic aberrations. We focused our analysis on those cancer types with the most considerable data on SALL family members.

### 5.1. Breast Cancer

Breast cancer is the most diagnosed tumor and the leading cause of cancer death among women worldwide. Thus, understanding its molecular mechanisms has become necessary for early diagnosis and successful treatment. The intrinsic classification of breast cancer distinguishes four subtypes: luminal A, luminal B, basal-like, and HER2-positive. This classification allows for the development of specific types of clinical management for breast cancer patients. In this line, the expression of the estrogen receptor (ER), progesterone receptor (PR), and HER2 biomarkers are related to a better prognosis and response to therapy in breast cancer patients than those with the absence of these biomarkers [135]. Several independent studies suggest that SALL1, SALL2, and SALL4 play a role in the origin, progression, and response to breast cancer therapy.

SALL2 is a putative target gene of MYB, a transcription factor predicted as a prognostic gene signature across molecular breast cancer subtypes. More importantly, MYB and SALL2 were suggested to attenuate histological grade promotion and prevent breast cancer progression [136]. In addition, a differentially weighted graphical LASSO analysis showed *SALL2* to be among the top 10 genes that are highly relevant in studies on the discovery of breast cancer biomarkers [137]. In addition, a transcription profiling analysis identified SALL2 to have been significantly reduced during tamoxifen therapy by a mechanism that involves the hypermethylation of the *SALL2* promoter. SALL2 transcriptionally upregulates estrogen receptor-alfa (ESR1) and PTEN, and directly binds to their promoters. Accordingly, the depletion of SALL2 decreased ESR1 and PTEN expression, activated the Akt/mTOR signaling, and resulted in estrogen-independent growth and tamoxifen resistance in ERα-positive breast cancer. Of relevance here is 5-azacitidine, a DNMTi (DNA methyltransferase inhibitor), which triggered SALL2 restoration and sensitized tamoxifen-resistant breast cancer to tamoxifen therapy in vivo [54]. Consistent with the negative regulation of cell proliferation by SALL2 [26,38], bioinformatic studies using public data and R2 software showed a negative correlation between *SALL2* and *CCNE1* in breast cancer samples [26]. The studies above suggest that SALL2 behaves as a tumor suppressor in breast cancer.

Similar to SALL2, SALL1 is downregulated in human breast cancer cells and tissues and correlates with ESR1 expression. An immunohistochemistry analysis of 17 tissues indicated that the number of SALL1 positive cells is significantly higher in estrogen receptor-positive (ER+) than in estrogen receptor-negative (ER−) patients [36]. Five-azacitidine also restored SALL1 expression on methylated breast cancer cell lines [92]. Consistent with its tumor suppressor role in breast cancer, SALL1 inhibited tumor growth, metastasis in the lung and liver, and promoted cell cycle arrest and senescence. Mechanistically, SALL1-dependent growth inhibition and senescence involved the recruitment of the NuRD complex and the activation of the p38 MAPK, ERK1/2, and mTOR signaling pathways [36]. Similarly, SALL1 knockdown led to the differential expression of EMT markers, such as E-cadherin and vimentin in SUM149 breast cancer cells, leading to increased cell migration in vitro. Moreover, SALL1-knockdown SUM149 cells *xenografted* into immunodeficient NSG mice resulted in a significantly decreased tumor-free survival [138].

Unlike SALL1 and SALL2, studies showed higher SALL4 expression in breast cancer cell lines and primary tissues than their non-tumoral counterparts [45,90,139]. This high expression of SALL4 positively correlated with tumor size and lymphatic metastasis [45]. Functionally, SALL4 knockdown inhibited cell proliferation and induced the G0/G1 cell cycle arrest of MDA breast cancer cells, an effect explained by the SALL4 positive regulation of the Wnt/β-catenin pathway in breast tumors [45]. In addition, in vitro SALL4 knockdown decreased migratory cell ability and promoted focal adhesion dynamics by the negative regulation of EMT-related markers (ZEB1, vimentin, and integrin α6β1), which was directly associated with E-cadherin recovery [140,141]. Consistent with the previous results, in vivo studies indicated that vimentin regulation by SALL4 enhances EMT, mammosphere formation, and tumorigenicity. *SALL4* deficiency reduced lung colonization in MDA-MB-231 cells [142] and was associated with triple-negative (ER-, PR-, and HER2-) phenotypes [143]. Nevertheless, a recent study from 371 breast cancer patients showed that SALL4 expression positively correlates with PR protein level. PR was related to breast cancer stemness in vitro, similar to SALL4 [142]. For an extensive review on SALL4 and breast cancer, refer to [143].

Altogether, studies suggest that in breast cancer, SALL1 and SALL4 are involved in migration, invasion, and EMT by regulating common targets such as E-cadherin and vimentin, but in an opposite manner. Moreover, SALL1 and SALL2 act as tumor suppressors and are associated with proliferation arrest, ESR1 expression, and good prognosis. SALL1, SALL2, and ESR1 expression analyses could help to categorize breast cancer patients who may benefit from combined therapies: tamoxifen and DNMTi. The restoration of SALL2 and SALL1 expression with DNMTi may directly impact breast cancer treatment, increasing tamoxifen sensitivity in tamoxifen-resistant breast cancers.

### 5.2. Brain Tumors

Glioma refers to a primary brain or spinal tumor that derives from the neuroglial stem or progenitor cells. Glioblastoma multiforme (GBM) is the most malignant and frequently occurring type of brain cancer. Despite advances in treatment approaches, it remains incurable [144]. To date, all studies regarding the role of SALL proteins in brain tumors are focused on GBM.

Unlike its tumor suppressor role in breast and ovarian cancer [4,38,145,146], SALL2 is part of the four-core neurodevelopmental transcription factors (including POU3F2, SOX2, and OLIG2) in GBM, which are sufficient to fully reprogram differentiated glioblastoma cells (DGCs) into stem-like tumor propagating cells (TPCs). In this context, SALL2 binds and activates TPC-specific regulatory elements. However, its interaction with the DNA is not associated with the previously CG-rich consensus motif identified by Gu and collaborators, found in cell cycle and pro-apoptotic gene promoters [49,65]. Instead, in the GBM context, SALL2 interacts with SOX2, which could explain the binding of SALL2 to SOX-like AT-rich elements [6]. Loss of function experiments through the induction of the miR-302/367 cluster in U87 cells revealed the importance of SALL2 and the other three transcription factors in brain tumor malignancy [101]. It was further supported by using lipopolymeric nanoparticle-containing combo siRNA (OLIG2, POU3F2, SALL2, and SOX2) targeting brain tumor-initiating cells (BTICs) in a mouse brain tumor model. The combo siRNA downregulated the PI3K/AKT and the STAT3 signaling, decreasing tumorigenicity and providing survival benefits [147]. These results opened possible avenues for future therapies that target brain tumor-initiating cells.

SALL4 has also been associated with increased tumorigenicity due to higher expression levels in GBM human samples as compared with normal brain tissue. The analysis of 524 GBM patients from the TCGA database showed a robust negative correlation between SALL4 expression and overall survival [148]. Furthermore, SALL4 silencing in the U87 and U251 GBM cell lines reduced cell proliferation by increasing PTEN expression and decreasing the PI3K/AKT pathway activity, triggering G1 cell cycle arrest [44].

On the other hand, SALL1 downregulation has been described in cerebral glioma tissues, correlating with higher tumor grade and lower survival. The overexpression of SALL1 in the U87 and U251 GBM cell lines induced a decrease in the expression of β-catenin, c-MYC, cyclin D1, and EMT markers, impairing migratory and invasive cell capacity. It also increased p21 and p27 expression, leading to G0/G1 cell cycle arrest [149].

In summary, SALL1 functions as a putative tumor suppressor in GBM, but SALL2 and SALL4 are crucial for tumor development and progression, which is associated with PI3K/AKT pathway activation and the maintenance of an undifferentiated aggressive phenotype.

### 5.3. Blood Cancers

Hematological malignancies correspond to a heterogeneous group of lymphoid and myeloid neoplasms caused by the deregulation of normal hematopoietic processes. They include leukemias, lymphoma, and multiple myelomas and are classified based on genetics, immunophenotype, morphology, and molecular and clinical characteristics [150]. SALL1, SALL2, and SALL4 were associated with hematological malignancies.

Two independent genome-wide analyses confirmed the hypermethylation of the aberrant SALL1 promoter in human samples of acute lymphocytic leukemia (ALL) and chronic lymphocytic leukemia (CLL). *SALL1* hypermethylation correlated with worse overall survival in patients [151,152]. Furthermore, SALL1 seems to have an oncogenic role in acute myeloid leukemia (AML). SALL1 silencing decreased human primary AML cell proliferation, resulted in low AML engraftment into NSG mice, and correlated with the upregulation of PTEN and the downregulation of mTOR, β-catenin, and NF-қB expression. This oncogenic role of SALL1 in AML may be associated with the expression of SALL1 isoform 2, characterized by the loss of exon 1 (Figure 1). SALL1 isoform 2 is expressed in AML but not in normal bone marrow, suggesting that this isoform has a different behavior than other SALL1 isoforms [9]. Thus, depending on the hematological malignancy, SALL1 could play different roles, as a tumor suppressor in lymphoid progenitor-derived leukemia (ALL-CLL) and as an oncogene in myeloid progenitor-derived leukemia (AML). Identifying the SALL1 isoform deregulated in ALL and CLL could help define the role and mechanisms of SALL1 in leukemia.

SALL2 is expressed in normal bone marrow but absent or weak in AML samples, suggesting that lost or reduced SALL2 expression is related to myeloid leukemogenesis. Moreover, *Sall2^−/−^* or *Sall2^−/+^*/*p53^−/−^* mice showed enhanced tumorigenesis, lymphoma progression, metastasis, and mortality rate compared with the *Sall2^+/+^*/*p53^−/−^* mice [25]. Interestingly, Histone Deacetylase Inhibitor (HDACi) TSA treatment induced SALL2-dependent apoptosis in Jurkat T cells. The genomic deletion of SALL2 suppressed PARP cleavage, decreased apoptotic cell population, and increased cell viability under TSA treatment. In contrast, the overexpression of SALL2 decreased cell viability. These data indicate that SALL2 is required for the apoptotic response in Jurkat T cells, an acute T cell leukemia model, thus supporting its role as a tumor suppressor [51].

SALL4 is highly expressed in hematopoietic malignancies and is associated with deteriorated disease status in patients [153]. Three SALL4 functional isoforms were identified in blood cancers in humans and mice (A, B, and C). Most studies are on the A and B variants (Figure 1). SALL4A is the full-length isoform, and SALL4B lacks exon 2 [12]. In transgenic mouse model studies, the SALL4A and SALL4B isoforms bound β-catenin, synergistically activating the WNT/β-catenin pathway, which plays a critical role in controlling leukemia stem cell self-renewal [154]. SALL4 also promoted leukemogenesis by repressing the tumor suppressor PTEN, similar to its breast cancer function. The mechanism involves the interaction of SALL4 with the histone deacetylase (HDAC) NuRD complex. Blocking SALL4/HDAC interaction with a peptide derived from the amino-terminal 12-amino-acid sequence of SALL4 led to higher PTEN expression and an antiproliferative effect on SALL4-expressing cancer cells. Of relevance here is that similar to SALL4 down-regulation, the SALL4 peptide treatment of primary AML cells impaired leukemic engraftment in vivo [155].

Together, these studies suggest that SALL1 and SALL2 play opposite roles in myeloid progenitor-derived leukemia, as oncogene and tumor suppressor, respectively. However, in lymphoid progenitor-derived leukemia (ALL-CLL), SALL1 could act as a tumor suppressor. SALL4 is oncogenic, required for leukemia stem cell self-renewal. Remarkably, both SALL1 and SALL4 positively regulate the WNT/β-catenin pathway. For a more comprehensive overview of SALL4 research on blood cancers, refer to [153].

### 5.4. Colorectal Cancer

Colorectal cancer (CRC) is the second most diagnosed malignancy worldwide in women and the third in men [156]. Advances in CRC research have increased the disease treatment options. Despite new approaches in the treatment of CRC patients, none is entirely effective, resulting in high recurrence even after tumor resection. Thus, it is necessary to search for new targets to facilitate the diagnosis and treatment [156]. In this context, SALL proteins could have a promising role.

The epigenetic inactivation of *SALL1* in epithelial cancers, including CRC, was identified using the MIRA (methylated-CpG island recovery assay) and CpG island arrays. *SALL1* promoter methylation was present in 83% of CRC and 89% of adenomas, while methylation frequency in normal tissues was 38% [92]. Additionally, Zhang and collaborators identified *SALL1* and *SALL3* as part of a high-risk group of genes differentially expressed within a vast number of genes whose methylation status also differed when comparing tumor and adjacent normal tissue. The study suggested *SALL1* and *SALL3* as being the new candidate biomarkers of poor prognosis in CRC. Epigenetic mechanisms partly mediate the loss of *SALL1* and *SALL3* in CRC [157]. Recently, *SALL1* was identified as one of nine prognostic gene signatures predicting survival in CRC patients [158].

There are some conflicting studies on the expression of SALL4 in CRC. SALL4 is upregulated, and its expression positively correlates with tumor stage, metastasis to lymph nodes, and poor differentiation in CRC samples [159,160]. Accordingly, a Kaplan–Meier analysis conducted after five years of follow-up on 135 CRC patients associated SALL4 expression with a lower survival rate as compared to the SALL4 negative group [161]. However, an immunohistochemical analysis of 149 patients revealed the significantly lower expression of SALL4 in CRC (46.3%) than in atypical hyperplasia (68.0%) and normal tissue (78.9%) [162]. However, as in previous studies, SALL4 expression positively correlated with lymph node metastasis, tumor node metastasis, and Dukes’ stages. SALL4 and β-catenin positively correlated in CRC tissues and cells, showing co-localization and interaction [162]. This study suggested that the function of SALL4 in promoting lymph node metastasis and the advanced clinical stage is partly due to its interaction with β-catenin. In addition, the flavonoid chrysin treatment of CT29 murine CRC cells decreased SALL4 expression, resulting in apoptosis induction associated with increased BAX levels and caspase 3/9 activity and with a decrease in tumor size in allograft assays [163]. The upregulation of SALL4 in CRC is partly due to the low aberrant expression of miR-3622a-3p, one of the SALL4 upstream negative regulators (Table 2). The overexpression of miR-3622a-3p in CRC cells decreased stemness features and EMT-related markers via the targeting of SALL4 [110]. In addition, miR-219-5p also inhibited colon cancer carcinogenesis by targeting SALL4 [109].

There are no functional studies of SALL2 or SALL3 in CRC; however, loss of heterozygosity (LOH) has been reported in the 14q12-13 region, a chromosomal region where the *SALL2* gene is located [72]. Bioinformatic analysis of massive CRC public data indicates that SALL2 is significantly downregulated in CRC [6]. However, another recent bioinformatic study identified a positive correlation between SALL2 and the degree of tumor-stromal cell infiltrates in colon and rectum adenocarcinomas [164].

Interestingly, in a small-sample-size study of primary and metastatic tumors from four patients with CRC, SALL3 was identified as one of seventeen genes significantly upregulated in CD133+ cells as compared to CD133− CRC cells, suggesting its potential as a biomarker of CRC stemness [165]. Thus, SALL proteins can positively or negatively affect CRC progression by different pathways and mechanisms. However, additional studies are required to elucidate the role of the SALL proteins in CRC.

### 5.5. Hepatocellular Carcinoma

Hepatocellular carcinoma (HCC) is the most common primary liver cancer worldwide. Carcinogenesis is associated with alcohol consumption, smoking, and genetic background, but the primary cause is the hepatitis B virus (HBV) infection. A recent meta-analysis proposed an association between HBV infection and gene methylation in HCC development [166]. SALL3 tumor suppressor function was associated with CpG island methylation in HCC. SALL3 directly interacts with DNMT3A, decreasing CpG methylation status [167]. Thus, the epigenetic silencing of SALL3 increases DNMT3A binding to chromatin, resulting in aberrant CpG island methylation that should contribute to HCC development [168,169].

Contrary to SALL3 hypermethylation, the SALL4 promoter is hypomethylated in HCC tissues and cell lines infected with HBV and the hepatitis C virus (HCV), which leads to the aberrant overexpression of the SALL4A and SALL4B isoforms [170]. Several studies demonstrated that the high expression of SALL4 in HCC is related to stem cell characteristics and poor prognosis [171,172,173,174,175]. SALL4 was detected in HCC, but its expression was weak in benign lesions and undetected in adjacent noncancerous hepatic tissues. Moreover, SALL4 was associated with α-fetoprotein (AFP), a well-known diagnostic marker in HCC. The detection of SALL4 combined with AFP may be helpful for prognostic stratification. Patients with higher levels of SALL4 and AFP were associated with a worse prognosis [172]. Additionally, a recent investigation suggests that a quantitative analysis of the liver function by contrast-enhanced ultrasonography (CEUS) in conjunction with a quantitative analysis of the SALL4/Wnt/β-catenin axis expression may serve as an early diagnosis method for HCC patients [176].

A recent study proposed that the inflammatory microenvironment promotes stemness and the metastatic phenotype in HCC via the NF-κB/miR-497/SALL4 axis. TNF-α activated NF-κB and repressed the miR-497 promoter. Consequently, the upregulation of SALL4 promoted stem cell self-renewal and metastasis [177]. On the other hand, SALL4 could impact the tumor microenvironment and support tumor progression by affecting the HCC exosome content. SALL4 bound to the miR-146a-5p promoter and positively regulated its expression in HCC exosomes. The miR-146a-5p from exosomes increased the number of M2-polarized tumor-associated macrophages that support tumor progression [178].

Altogether, the evidence points to SALL4 as a potential candidate for HCC therapy. Additionally, SALL3’s relationship with global methylation appears to be an attractive field of study. Demethylating agents are increasingly being used and proposed for clinical trials in different tumors, including HCC [179].

### 5.6. HNSCC and Cervical Cancer

Strong evidence associates the human papillomavirus (HPV) with several human diseases, including cancer. To date, more than 170 HPV genotypes have been described, of which the high-risk HPV (HR-HPV) types 16, 18, 31, 33, and 35 are related to malignancies such as cervical carcinoma and head and neck squamous cell carcinoma (HNSCC). HNSCC includes a heterogeneous group of malignancies that arise in the oral cavity, pharynx, or larynx [180].

The first study that associated SALL proteins with HPV-related cancer showed that oncoprotein E6 from HR-HPV infection binds SALL2 and induces its stabilization in cervical cancer cells. This interaction prevented the binding of SALL2 to the p21 promoter, leading to the accumulation of inactive SALL2 in SiHa, Caski, and HeLa HR-HPV positive cervical cancer cell lines [146]. However, the cellular location and significance of the inactive SALL2 protein in cervical and other HPV-related cancers require further investigation.

Subsequent evidence of a relation between SALLs and HPV-related cancers conveys their epigenetic regulation. The hypermethylation of the *SALL3* promoter was described in different HPV-related cancer cell lines and tissues such as cervical cancer and HNSCC [95,96,181]. *SALL3* hypermethylation correlates with reduced disease-free survival (DFS) in stage III and IV HNSCC patients [96]. Moreover, the *SALL3* chromosomal locus is part of a region with genes identified as significant prognostic biomarkers in HNSCC patients. Here, LOH on chromosome 18q23 is associated with significantly decreased survival in HNSCC patients [95]. Similar to *SALL3*, *SALL1* is aberrantly hypermethylated in HNSCC. It correlates with reduced DFS in early-stage T1 and T2 patients and with the methylation status of the *SALL3* promoter [97]. In the same context, *SALL2* hypermethylation correlated with the methylation status of *SALL1* and *SALL3*. Thus, more than one *SALL* (*SALL1*–*3*) hypermethylation positively correlates with a worse prognosis and lower DFS in HNSCC [94]. In contrast to SALL1 and SALL3, high SALL4 expression is correlated with disease recurrence and decreased DFS rates in HNSCC [121]. SALL4 expression increased G1 to the S-phase cell cycle progression in cervical cancer cell lines by a mechanism that involves the increment of β-catenin expression, a necessary cofactor for activating the Wnt/β-catenin signaling pathway involved in cell proliferation [41].

In summary, members of the SALL family are related to HNSCC and cervical carcinoma, with a loss of function of *SALL1*–*3* by epigenetic silencing. Furthermore, SALL4 upregulation is associated with a poor prognosis. The involvement of SALL proteins in other HPV-related cancers such as penile, vulvar, vaginal, or anal cancer awaits investigation.

## 6. Targeting SALLs for Cancer Therapy

Considering the evidence on the role of the SALL family members in cancer initiation and progression, targeting SALLs provides a unique therapeutic opportunity for cancer treatments.

Even though transcription factors have remained challenging drug targets, several in vitro and in vivo approaches targeting SALL4 protein activity have already been investigated.

The HDAC-1 and -3 inhibitor Entinostat was identified as a potential treatment for SALL4-expressing cancers. The study used a panel of 17 lung cancer cell lines with varied SALL4 expression levels, showing that cells expressing high levels of SALL4 were more sensitive to Entinostat treatment [182]. However, HDAC inhibitors are not selectively targeting SALL4 expressing cells. Pharmacological peptides that exclusively target cancer cells expressing SALL4 were tested as potential cancer therapeutic agents. A 12-amino-acid peptide that disrupts the interaction between SALL4 and the NuRD complex comprising HDAC1 and HDAC2 was tested in AML and HCC. The peptide disrupted the interaction between SALL4 and HDAC, which blocked the NuRD-mediated SALL4 repression function [155,174]. Similarly, another peptide, known as PEN-FFW, was recently designed to target SALL4 in HCC cell lines. The peptide disrupted the SALL4–NuRD interaction via the blocking of the SALL4 interaction with RBBp4, specifically inhibiting the transcription-repressor function of SALL4. Treatment of HCC cells with the PEN-FFW peptide induced apoptosis, enhanced cell adhesion, and dramatically inhibited xenograft tumor growth [183]. The use of miRNAs targeting SALL4-associated HCC has also been proposed. Let-7/miR-98-induced SALL4 depletion decreased the expression of MMP2/9, Fibronectin, *n*-cadherin, and increased E-cadherin, which correlated with reduced migration/invasion and EMT in an HCC in vivo cancer model [100].

Alternative strategies are the use of drugs that can induce the degradation of SALL4. Small immunomodulatory drugs (IMiDs), such as thalidomide and derivatives, induce ubiquitination and the proteasomal degradation of zinc finger transcription factors. The mechanism involves recruiting C2H2 zinc finger (ZnF) domains to Cereblon (CRBN), the substrate receptor of the CRL4CRBN E3 ubiquitin ligase [184]. Thalidomide induced the robust degradation of SALL4 in the neuroblastoma (Kelly and SK-N-DZ) cell lines and the MM1 multiple myeloma cell line [185]. However, because of the critical role of SALL4 in limb development, treatment with thalidomide or analogous drugs during pregnancy could contribute to severe birth developmental abnormalities [17,185,186,187].

Studies on the reestablishment of the SALL tumor suppressor function have focused on demethylating agents’ potential use. As indicated previously, *SALL1*, *SALL2*, and *SALL3* hypermethylation are associated with bad prognosis in several cancers [54,55,92,93,94,95,96,97], and the epigenetic silencing of SALL2 confers tamoxifen resistance in breast cancer [54]. Treatment with 5-Aza-dC, a DNMT inhibitor, increased the sensitivity of SALL2 hypermethylated breast cancer to tamoxifen therapy in vitro and in vivo [54]. This result suggests that treatment with DNMT inhibitors might overcome tamoxifen resistance in breast cancer. However, co-therapy, including both a DNMT inhibitor and tamoxifen, might be an appropriate therapy for a subset of patients with breast cancer. In addition, Histone deacetylase inhibitors, including TSA (Class I and II inhibitor), Panobinostat (Pan-HDAC inhibitor), Vorinostat/SAHA (class I and II inhibitor), and Chidamide (class I inhibitor), also upregulated SALL2 in Jurkat T cells by a mechanism involving the recruitment of Sp1 and p300 to the P2 promoter [51].

Additional therapeutic strategies to restore tumor suppressor function could be related to the modulation of SALL1 or SALL2 protein levels by altering their ubiquitylation and subsequent proteasome degradation, or by inhibiting still unknown negative regulators of the transcription factor expression. Thus, efforts should also focus on identifying novel SALL regulators and partners.

## 7. Concluding Remarks/Future Perspectives

Increasing evidence shows an association between SALL family members with human cancers. Consistently, SALL4 acts as an oncogene; however, SALL1–3 play dual roles, depending on the cancer context and disease stage. Still, most studies support the tumor suppressor role of SALL1 and SALL2.

As expected with regard to the essential role of transcription factors in gene expression regulation, all SALLs directly or indirectly impact the hallmarks of cancer, including cell proliferation, cell migration, and cell survival. An analysis of the current literature highlights some common signaling pathways between SALL members such as PI3K/Akt and Wnt/β-catenin and common target genes such as *PTEN*, *CCDN1*, *c-MYC*, *VIM*, and *CDH1*. SALL proteins interact with specific protein partners to perform their functions; however, as indicated here, some partners are shared among family members. Examples are β-catenin, a partner of SALL1 and SALL4, and DNMT3, a partner of SALL3 and SALL4. In addition, all four SALLs interact with SOX2 in the stemness context [63,65,66,67], and with the NuRD complex that is associated with transcriptional repression activities [5,28,32,34,36].

According to these few examples of shared signaling pathways and interactors related to the function of SALL proteins, genetic analysis of specific cancer types should consider all the family members. Specific SALLs might act oppositely, activating or repressing the same pathway, inducing or repressing the target gene. In addition, changes in expression of a particular SALL member, the presence of cancer-related isoforms, or specific mutations might affect the relative availability of shared protein partners and consequences as they could compete for interaction. Thus, the expression levels of each SALL family member and the pattern of interactors expressed in specific cells might influence cellular outcomes. These aspects are relevant as deregulated SALL1–4 have been reported in at least 11 cancer types analyzed in this review (Figure 3, Table 2).

Current literature indicates that SALL1 and SALL2 might have a dual role in cancer. A recent study using a database search and a literature annotation of 12 main cancers identified genes with oncogenic and tumor suppressor functions, which are called double agents [188]. Most of them encoded transcription factors or kinases and exhibited dual biological functions. Still, double agents mainly function as tumor suppressors in normal tissues. The dual role of SALL1 and SALL2 in cancer might relate to the expression of specific isoforms, the cancer stage, and the context (primary tumor versus metastasis), and/or the presence of particular partners. As an example of this, unlike its proposed tumor suppressor role in several cancer types, SALL2 acts as a cancer-promoting factor in GBM. In this context, SALL2 interacts with SOX2, and this interaction may lead to the binding of SALL2 to DNA elements (AT-rich) different from the SALL2 consensus sequence (GC-rich) [6,65], likely explaining its oncogenic role in GBM.

Another significant finding is the common epigenetic regulation of *SALL* genes and the association of epigenetic marks with patient prognosis. In general, the hypermethylation of *SALL1*–*3* is associated with a bad prognosis in several cancers. Relevant for cancer therapeutics is that the demethylating agent 5-azacytidine increases SALL1 and SALL2 expression and is associated with a better prognosis. This evidence suggests that the *SALL* status is a valuable tool for predicting the response to chemotherapeutic drugs and other cancer treatments in various malignancies [54,55,92,94,95,96,97,145,168,181,189]. On the contrary, *SALL4* promoter hypomethylation and miRNAs-dependent SALL4 regulation are common events in several cancer types [98,99,100]. Likewise, studies indicate that the oncogenic function of SALL4 and the tumor suppressor function of SALL1 greatly rely on the recruitment of the NuRD complex. Targeting the SALL4/NuRD interaction holds great potential for cancer treatment (see the section above on Targeting SALLs for cancer therapy). Although SALL1 is frequently epigenetically inactivated, it would be interesting to investigate the existence of mutations that impair its association with NuRD. Future studies should continue to focus on identifying the subunits of NuRD and target genes recruited by different SALL proteins in cancer.

Thus, efforts should pursue the identification of novel SALL regulators, partners, and targets. Additional mechanistic and comprehensive studies, including more than one SALL protein in similar cancer contexts, are required to understand the significance of SALLs’ alterations in cancer. In this regard, a recent article used the TCGA pan-cancer data and NCI-60 database to conduct a comprehensive analysis of *SALL* genes. The study suggests that SALLs associate with immune infiltrate subtypes, with a close association between different degrees of stromal and immune cell infiltration. Furthermore, it supports the idea that *SALLs* are related to cancer cell resistance [164]. However, further functional studies of *SALL* genes are required to confirm these findings.

## Figures and Tables

**Figure 1 cancers-13-06292-f001:**
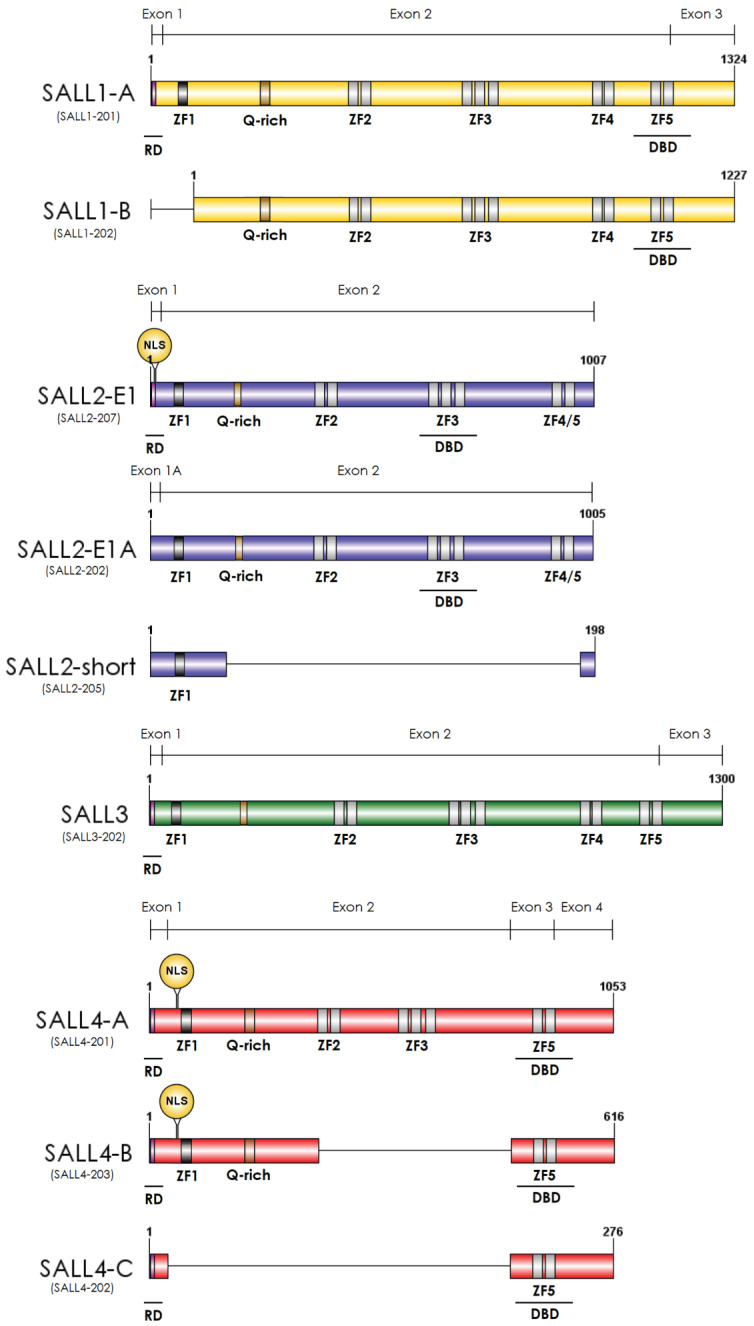
Schematic representation of the main SALL protein isoforms. The colors represent the different SALL proteins; yellow, blue, green, and red for SALL1, SALL2, SALL3, and SALL4, respectively. Dark grey rectangles at the N-terminal region represent the C2HC-type Zinc Finger Motif (ZF1). Light grey rectangles represent C2H2-type Zinc Finger Motifs 2–5 (ZF2–ZF5); all of them are in SALL1-A, SALL1-B, and SALL3. SALL4 lacks ZF4, and SALL2 has a motif that differs from the others, located between ZF4 and ZF5 (depicted as ZF4/5). The pink rectangle at the N-terminal region represents the conserved 12-amino-acid region that binds to the NuRD complex, named repression domain (RD). The RD is in SALL1A, SALL2 E1, SALL3, SALL4 A, B, and C. The yellow rectangle between ZF1 and ZF2 corresponds to the conserved Glutamine-rich (Q-rich) region. The circle at the N-terminal region shows the nuclear localization sequences (NLS) described only for SALL2 E1A, and SALL4 A and B. The ensembl transcript ID is under each isoform name in brackets. Exon representation is above each SALL protein, and the protein length is at the end of each isoform. ZF: Zinc Finger; Q-rich: Glutamine-rich; RD: Repression Domain; DBD: DNA Binding Domain; NLS: Nuclear Localization Sequence.

**Figure 2 cancers-13-06292-f002:**
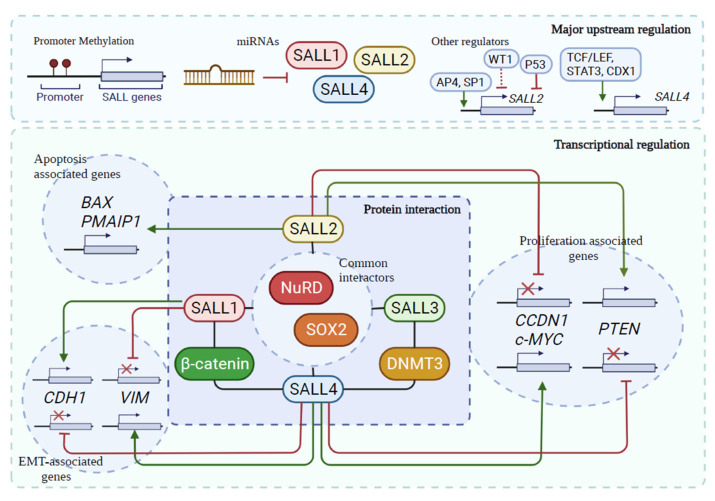
Common upstream regulation, partners, genes, and cellular functions of the SALL family. Epigenetic changes, including gene hypermethylation, hypomethylation, and miRs, are common mechanisms of *SALL* regulation. *SALL1–3* promoters are hypermethylated in several cancers [54,55,92,93,94,95,96,97]. In contrast, the *SALL4* promoter is hypomethylated in AML and MDS [98,99] and regulated by miRNAs in multiple cancer types [100]. Additionally, SALL1 and SALL2 are regulated by miRs (Table 1). Regulation by specific transcription factors depicted for *SALL2* and *SALL4*. SALL proteins interact with specific partners to perform their functions; shared protein partners among the family include β-catenin and DNMT3. In addition, the four SALLs interact with the NuRD complex and with SOX2. Common transcriptional targets of SALLs are associated with cell proliferation and migration/invasion. SALL2 and SALL4 oppositely regulate *CCDN1*, *c-MYC*, and *PTEN*. Similarly, SALL1 and SALL4 oppositely regulate *CDH1* (E-cadherin) and *VIM* (vimentin). However, SALL1 and SALL4 are both negative regulators of PTEN. Moreover, apoptosis-associated genes, such as *BAX* and *PMAIP1* (*NOXA*), are regulated by SALL2. Green lines: positive regulation. Red lines: negative regulation. Dotted lines: proposed association.

**Figure 3 cancers-13-06292-f003:**
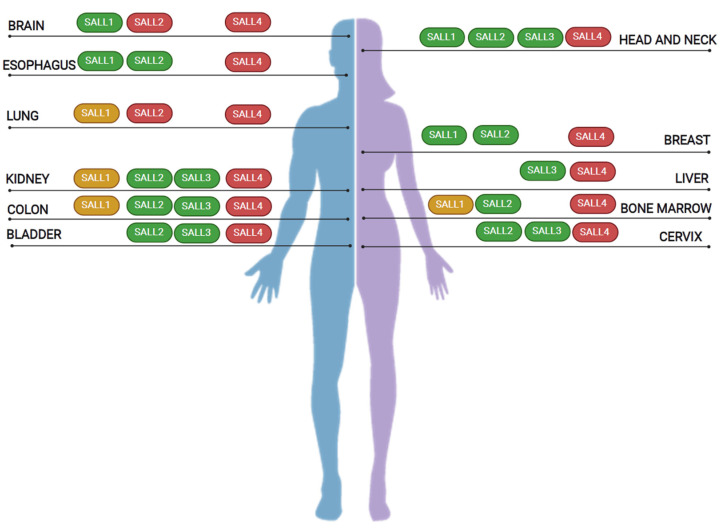
SALL proteins in cancer. SALL proteins are deregulated in major cancer types, including lung, colon, and breast cancers. As shown above, independent studies identified alterations in more than one family member in specific cancer types. According to genetic alterations, isoform expression, and changes in their expression, they are classified as oncogenes (red), tumor suppressors (green), or genes with a dual role in cancer (yellow).

**Table 1 cancers-13-06292-t001:** Summary of all microRNAs known to regulate SALLs.

Cancer Type/Cellular Model	microRNA	Target	SALL Status/Key Findings	Experimental Approach	Ref.
Glioma/Glioblastoma	miR-302/367 cluster	SALL2	miR-302/367 cluster can reprogram tumor cells, generating more benign phenotypes by suppressing OCT3/4, SOX2, KLF4, c-MYC, POU3F2, OLIG2, and SALL2	qRT-PCR, cytokine array analysis	[101]
Glioma/Glioblastoma	miR-16	SALL4	miR-16 inhibits proliferation, migration, and invasion in glioma cells by directly targeting SALL4	qRT-PCR and Luciferase reporter assay	[102]
Glioma/Glioblastoma	miR-103/miR-195/miR-15-B	SALL4	miR-103, miR-195, and miR-15-B inhibit proliferation, migration, and invasion and promote apoptosis in glioma by directly targeting SALL4	qRT-PCR, Western blot, and Luciferase reporter assay	[103]
Glioma/Glioblastoma	miR-107	SALL4	miR-107 inhibits proliferation and promotes apoptosis in glioma cells by directly targeting SALL4	qRT-PCR, Western blot, and Luciferase reporter assay	[104]
Glioma/Glioblastoma	miR-181b	SALL4	miR-181b inhibits proliferation, migration, and invasion and promotes apoptosis in glioma by directly targeting SALL4	qRT-PCR, Western blot, and Luciferase reporter assay	[105]
Gastric cancer	miR188-5p	SALL4	miR-188-5p promotes proliferation and migration by upregulating SALL4 expression, nuclear translocation, and transcription	qRT-PCR, Western blot, and Luciferase reporter assay	[106]
Gastric cancer	miR-16	SALL4	miR-16 inhibits proliferation and migration in GC by directly targeting SALL4	qRT-PCR and Luciferase reporter assay	[107]
Colorectal cancer	miR-181a-2 *	SALL1	miR-181a-2 * correlates with a trend of repression of SALL1 and high methylation status of the *SALL1* promoter	qRT-PCR and bisulfite modification followed by quantitative methylation- specific PCR (qMSP)	[108]
Colorectal cancer	miR-219-5p	SALL4	miR-219-5p inhibits proliferation, migration, and invasion, reduces drug resistance, and promotes apoptosis in CRC by directly targeting SALL4	qRT-PCR, Western blot, and Luciferase reporter assay	[109]
Colorectal cancer	miR-3622a-3p	SALL4	miR-3622a-3p inhibits proliferation, cell cycle, migration, invasion, and stemness features and promotes apoptosis by targeting SALL4	qRT-PCR, Luciferase assay, RNA immunoprecipitation (RIP) assay, and pull-down assay	[110]
Embryonic stem cell	miR15-B	SALL4	Anti-miR-15b enhances expansion of HSC in vitro by targeting SALL4	qRT-PCR	[111]
Embryonic stem cell	miR-294 and let-7 miRNAs	SALL4	Let-7 miR family inhibits self-renewal genes, and miR-294 indirectly induces self-renewal genes, including SALL4	qRT-PCR, Western blot, and Luciferase reporter assay	[112]
Oral squamous cell carcinoma	miR-103	SALL4	miR-103 inhibits cell proliferation and invasion by downregulating *SALL4* mRNA in Tca8113 cells	Luciferase reporter assay	[113]
Breast cancer	SNHG12 and miR-15a-5p	SALL4	Long non-coding RNA (lncRNA) small nucleolar RNA host gene 12 (SNHG12) promotes proliferation, migration, and invasion and inhibits apoptosis in breast cancer by upregulating SALL4 expression via sponging miR-15a-5p; SALL4 is a direct target of miR-15a-5p	qRT-PCR, Western blot, and Luciferase reporter assay	[114]
Renal cell carcinoma	miR-942	SALL1	miR-942 affects survival of patients with renal cell carcinoma by negatively regulating the expression of SALL1	RNA-seq and qRT-PCR	[115]
Prostate cancer	miR-4286	SALL1	miR-4286 regulates proliferation and apoptosis in PCa cells by directly targeting the 3′UTR of *SALL1* mRNA	qRT-PCR and Luciferase reporter assay	[116]
Lung cancer	HOXA11-AS and miR-3619-5p	SALL4	lncRNA homeobox A11 antisense (HOXA11-AS) promotes proliferation, migration, invasion, and glycolysis in non-small cell lung cancer (NSCLC) cells by upregulating SALL4 expression via sponging miR-3619-5p; SALL4 is a direct target of miR-3619-5p	qRT-PCR, Western blot, and Luciferase reporter assay	[117]
Osteosarcoma	ZEB2-AS1 and miR-107	SALL4	lncRNA ZEB2-AS1 (ZEB2-AS1) promotes proliferation, invasion, and metastasis and inhibits apoptosis in osteosarcoma cells by upregulating SALL4 expression via sponging miR-107; SALL4 is a direct target of miR-107	qRT-PCR, Luciferase assay, and RNA pull-down assay	[118]
Hepatocellular carcinoma	miR-296-5p	SALL4	miR-296-5p inhibits stemness potency of hepatocellular carcinoma (HCC) cells via the Brg1/Sall4 axis; Brg1 binds to the *SALL4* promoter	qRT-PCR, Western blot, Luciferase reporter assay, and Chromatin immunoprecipitation (ChIP) assay	[119]
Hepatocellular carcinoma	miR-15a	SALL4	Exosomal miR-15a reduces proliferation, migration, invasion, and survival by directly targeting SALL4	qRT-PCR, Western blot, and Luciferase reporter assay	[120]

**Table 2 cancers-13-06292-t002:** Deregulation of SALLs in other cancers.

Cancer Type	SALL Member	Expression Levels	Genetic Alteration/Regulation	Association With Cancer/Biological Process	Proposed Cancer Role	Ref.
Lung	SALL1	High	Undescribed	Expression correlated with lower overall survival of NSCLC patients	Oncogene	[122]
Lung	SALL2	Low	LOH	Undescribed	Undescribed	[71]
Lung	SALL4	High	Undescribed	Expressed in 88% of the lung cancer samples May be used as a diagnostic marker	Oncogene	[123]
Lung	SALL4	High	Undescribed	SALL4 knockdown inhibits cell proliferation by cell cycle arrest at the GO/G1 phase Loss of SALL4 function inhibits migration, invasion and reduces the transplanted tumors size in an in vivo model	Oncogene	[43]
Lung	SALL4	High	Undescribed	SALL4 silencing sensitizes cells to cisplatin, carboplatin, and paclitaxel treatment	Oncogene	[124]
Esophageal	SALL1	Low	Hypermethylation	*SALL1*, *ADHFE1*, *EOMES*, and *TFPI2* are proposed as part of a tumor suppressors panel with diagnostic relevance	Tumor suppressor	[93,125]
Esophageal	SALL2	Low in radioresistant ESCC cell lines	Hypermethylation	SALL2 overexpression enhances apoptosis after radiation and decreases migration, viability, and cisplatin resistance in TE-1/R and Eca-109/R cell lines	Tumor suppressor	[55]
Esophageal	SALL4	High	Undescribed	SALL4 silencing in ESCC cells is associated with suppressing cell migration, invasion, viability, and drug resistance in vivo SALL4 knockdown reduces epithelial-mesenchymal transition by targeting the Wnt/β-catenin signaling pathway	Oncogene	[42,126]
Bladder	SALL2	Low	LOH	Undescribed	Tumor suppressor	[70]
Bladder	SALL3	Low	Hypermethylation	*SALL**3, CFTR,* and *TWIST**1* are proposed as disease recurrence predictors	Tumor suppressor	[127,128]
Testicular tumors	SALL4	High	Undescribed	SALL4 is a novel sensitive and specific marker for testicular germ cell tumors	Oncogene	[129]
Kidney	SALL1	Low	miR-942	SALL1 inhibition plays a potential role in sunitinib resistance in RCC patients	Tumor suppressor	[115]
Wilms’ tumor	SALL1	High	Undescribed	Undescribed	Oncogene	[130,131]
Wilms’ tumor	SALL2	High	Undescribed	SALL2 was identified as one of the 27 signature genes highly expressed by comparing tumor samples with normal fetal kidneys	Oncogene	[132]
Kidney	SALL3	Low	Methylation	SALL3 downregulation may contribute to genome hypermethylation similar to VHL	Tumor suppressor	[133]
Wilms’ tumor	SALL4	High	Undescribed	Undescribed	Oncogene	[134]

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
