# Peer review of "SALL Proteins; Common and Antagonistic Roles in Cancer"

_cancers, 2021, doi:10.3390/cancers13246292_

Round 1
Reviewer 1 Report
This review by Alvarez et al. provides detailed insight into the role of SALL family transcription factors in cancer development. The gene family is briefly introduced and roles in cell functions contributing to tumorigenesis described. Regulatory mechanisms are discussed and specific roles in different cancer types are then covered in detail. Combined, it is an insightful review that will be of interest to the field and summarises a great deal of information from the literature. However, some aspects of the review should be improved and expanded.
- While the focus of the review is on role of SALL genes in cancer, it is insightful to compare known roles in mammalian development given that many developmental pathways are involved in cancer. Additional details on this area should be included as a short additional section.
- Related to the above point, mutations in SALL1 and SALL4 are found in developmental disorders. Given that the review discusses alterations in SALL activity in cancer, it would be helpful for the reader to briefly summarise what is known about SALL mutations and developmental disorders.
- The regulation of SALL proteins is discussed and focuses on promoter methylation and miRNA-dependent control. What is known about SALL gene regulation besides general promoter methylation? This should be discussed in the appropriate section.
- A link is drawn between SALL proteins and stemness as a hallmark of cancer development. However, this section (starting line 218) focuses on ESCs. What is known about the role of SALL genes in other stem cell types? For example, roles in kidney progenitors, HSCs and spermatogonial stem cells may be important to highlight. Do SALL genes play similar roles in different stem cell types?
- The interaction of SALL with the NURD co-repressor is briefly discussed. Do SALL proteins always promote target gene repression and what is the relevance of this interaction for cancer development? There is some evidence in ESCs that this interaction may not be relevant for gene regulation (PMID: 27471257) and this should also be highlighted.
- A number of cancers that involve SALL genes are discussed in detail. There seems to be an extensive number of publications detailing the role of SALL4 in hepatocellular carcinoma but this topic has not been selected for further discussion. It is recommended to discuss the tumor type in more detail with reference to some key studies (e.g. PMID: 23758232)
- SALL proteins are clearly involved extensively in cancer development. Does this provide unique therapeutic opportunities? This is an area that should be expanded on at the end of the review. For example, could pharmacologic peptides that disrupt interaction with NURD be useful in cancer therapy (PMID: 29976840)?
- Please check the manuscript carefully – some errors in grammar/spelling are noted.
Author Response
Reviewer 1
This review by Alvarez et al. provides detailed insight into the role of SALL family transcription factors in cancer development. The gene family is briefly introduced and roles in cell functions contributing to tumorigenesis described. Regulatory mechanisms are discussed and specific roles in different cancer types are then covered in detail. Combined, it is an insightful review that will be of interest to the field and summarises a great deal of information from the literature. However, some aspects of the review should be improved and expanded.
R: We sincerely appreciate reviewer 1 general comments.
Specific concerns:
1.- While the focus of the review is on role of SALL genes in cancer, it is insightful to compare known roles in mammalian development given that many developmental pathways are involved in cancer. Additional details on this area should be included as a short additional section.
R: We have included an additional section “Essential roles of SALL genes during development” about the role of SALLs in mammalian development, lanes 139-203.
2.- Related to the above point, mutations in SALL1 and SALL4 are found in developmental disorders. Given that the review discusses alterations in SALL activity in cancer, it would be helpful for the reader to briefly summarise what is known about SALL mutations and developmental disorders.
R: In the “Essential roles of SALL genes during development” section, we summarized what is known about SALL mutations and developmental disorders, lanes 141-149.
The regulation of SALL proteins is discussed and focuses on promoter methylation and miRNA-dependent control. What is known about SALL gene regulation besides general promoter methylation? This should be discussed in the appropriate section.
R: In the “Common regulatory mechanisms for SALL proteins in cancer” section, we have included information about the loss of heterozygosity of SALL1-3 in certain cancers, and specific transcriptional regulators of SALL2 and SALL4 (lanes 332-344). The transcription factors involved in SALL2 and SALL4 regulation are included in the new Figure 2.
4.-A link is drawn between SALL proteins and stemness as a hallmark of cancer development. However, this section (starting line 218) focuses on ESCs. What is known about the role of SALL genes in other stem cell types? For example, roles in kidney progenitors, HSCs and spermatogonial stem cells may be important to highlight. Do SALL genes play similar roles in different stem cell types?
R: The role of SALL genes in other stem cell types is now included in lanes 291-303.
5.- The interaction of SALL with the NURD co-repressor is briefly discussed. Do SALL proteins always promote target gene repression and what is the relevance of this interaction for cancer development? There is some evidence in ESCs that this interaction may not be relevant for gene regulation (PMID: 27471257) and this should also be highlighted.
R: SALL proteins do not always promote target gene repression. As an example, under DNA damage, SALL2 E1A induces expression of pro-apoptotic BAX and NOXA genes by direct binding to specific consensus sites in the promoters (Escobar, 2015). A special section about SALLs and the NURD co-repressor during development and cancer is included in lanes 168-196.
6.- A number of cancers that involve SALL genes are discussed in detail. There seems to be an extensive number of publications detailing the role of SALL4 in hepatocellular carcinoma but this topic has not been selected for further discussion. It is recommended to discuss the tumor type in more detail with reference to some key studies (e.g. PMID: 23758232)
R: A special “Hepatocellular carcinoma” section is included in lanes 628-665.
7.- SALL proteins are clearly involved extensively in cancer development. Does this provide unique therapeutic opportunities? This is an area that should be expanded on at the end of the review. For example, could pharmacologic peptides that disrupt interaction with NURD be useful in cancer therapy (PMID: 29976840)?
R: The role of SALL proteins in cancer is indeed providing unique therapeutic opportunities. We apologize for not expanding this subject in the original version. We now included the “Targeting SALLs for cancer therapy” section, where we discussed some studies using different therapeutic approaches, lanes 718-772.
8.- Please check the manuscript carefully – some errors in grammar/spelling are noted.
R: We checked our manuscript carefully.
Reviewer 2 Report
This a clear, informative, and well written review which will makes a clear contribution to the cancer literature.
While reading the review several questions came to mind and possibly the inclusion of their answers may help enhance the information that the review delivers and increase the number of citations in the future.
- What is the homology between the family members? Which genes are evolutionarily more similar? The authors mention four paralogs; which is (potentially) the original family member of these vertebrate genes? Obviously, this is not an evolutionary paper, however is there a particular domain that gives the tumor suppressor and oncogene function?
- Could the (generalized) impingement of the SALL family members on cell cycle and EMT be shown in a schematic diagram?
- Does the phenotype of any knockout mice correlate with the oncogene and tumor suppressor function of SALL proteins in certain organs?
- Although the final perspectives paragraph touched on this point, I was left slightly disappointed that this section did not offer more strategies in targeting SALL family for potential cancer treatments. miRNA targeting, specific gene deletion/reactivation, tumors that may benefit for methylation inhibitors etc
Author Response
This a clear, informative, and well written review which will makes a clear contribution to the cancer literature. While reading the review several questions came to mind and possibly the inclusion of their answers may help enhance the information that the review delivers and increase the number of citations in the future.
R: We sincerely appreciate reviewer 2 general comments.
Specifics concerns:
1.- What is the homology between the family members? Which genes are evolutionarily more similar? The authors mention four paralogs; which is (potentially) the original family member of these vertebrate genes? Obviously, this is not an evolutionary paper, however is there a particular domain that gives the tumor suppressor and oncogene function?
R: In lanes 53-58, we have included additional information about the homology between family members, which we previously investigated (Hermosilla,V. 2017). Concerning a particular domain for cancer-related functions, experimental data indicate that the conserved 12 AA N-terminal domain that interacts with the NuRD complex is required for the oncogenic function of SALL4 and the tumor suppressor function of SALL1 (lanes 825-827). Also, a recent study highlights the C2H2 zinc-finger cluster 4 (ZFC4) domain as essential for the SALL4-dependent regulation of AT-rich genes, promoting neuronal differentiation. The ZFC4 domain is also found in SALL1 and SALL3 but not in SALL2 (lanes 308-310; Partier, R. 2021).
2.- Could the (generalized) impingement of the SALL family members on cell cycle and EMT be shown in a schematic diagram?
R: We replace figure 2 with a new figure 2 that includes additional regulators and the SALL’s related functions such as proliferation, EMT, apoptosis.
3.- Does the phenotype of any knockout mice correlate with the oncogene and tumor suppressor function of SALL proteins in certain organs?
R: Because of their essential roles during development, Sall1, Sall3, and Sall4 KO are lethal. Sall2 KO did not die or develop spontaneous tumors. However, when crossed with tumor-susceptible mice (p53-/-), it exhibited significantly accelerated tumorigenesis, tumor progression, and mortality rate than the Sall2+/+/p53-/- mice. The Sall2-/- or Sal2-/+/p53-/- mice showed thymus T-cell lymphoma that metastasized to the liver, lung, kidney, marrow, peripheral blood, and central nervous system. Most of the Sall2+/+/p53-/- mice showed lymphoma limited to the thymus and adjacent organs, such as the lung (Chai, L. 2011). Also, supporting a tumor suppressor function, immortalized Sall2-/- MEFs showed enhanced growth rate, foci formation, and anchorage-independent growth (Lanes 160-167 and Hermosilla, V. 2018)
4.- Although the final perspectives paragraph touched on this point, I was left slightly disappointed that this section did not offer more strategies in targeting SALL family for potential cancer treatments. miRNA targeting, specific gene deletion/reactivation, tumors that may benefit from methylation inhibitors etc. 5.- SALL proteins are clearly involved extensively in cancer development. Does this provide unique therapeutic opportunities? This is an area that should be expanded on at the end of the review. For example, could pharmacologic peptides that disrupt interaction with NURD be useful in cancer therapy (PMID: 29976840)?
R: We apologize for not including a special therapeutic opportunities section initially. We now included the “Targeting SALLs for cancer therapy” section covering this relevant aspect of the SALL family in cancer (lanes 718-772). Indicated paper is cited.